# P2K1 Receptor, Heterotrimeric Gα Protein and CNGC2/4 Are Involved in Extracellular ATP-Promoted Ion Influx in the Pollen of *Arabidopsis thaliana*

**DOI:** 10.3390/plants10081743

**Published:** 2021-08-23

**Authors:** Yansheng Wu, Hongmin Yin, Xinyue Liu, Jiawei Xu, Baozhi Qin, Kaili Feng, Erfang Kang, Zhonglin Shang

**Affiliations:** 1Key Laboratory of Molecular & Cellular Biology of the Ministry of Education, College of Life Sciences, Hebei Normal University, Shijiazhuang 050024, China; wuysh@126.com (Y.W.); yinhongmin123456@163.com (H.Y.); liuxinyue2521@163.com (X.L.); 18832506165@163.com (J.X.); qinbaozhi2021@126.com (B.Q.); kailifeng201606@126.com (K.F.); 2College of Bioscience and Bioengineering, Xingtai University, Xingtai 054001, China

**Keywords:** extracellular ATP (eATP), *Arabidopsis thaliana*, pollen grain, ion influx, signaling

## Abstract

As an apoplastic signal, extracellular ATP (eATP) is involved in plant growth and development. eATP promotes tobacco pollen germination (PG) and pollen tube growth (PTG) by stimulating Ca^2+^ or K^+^ absorption. Nevertheless, the mechanisms underlying eATP-stimulated ion uptake and their role in PG and PTG are still unclear. Here, ATP addition was found to modulate PG and PTG in 34 plant species and showed a promoting effect in most of these species. Furthermore, by using *Arabidopsis thaliana* as a model, the role of several signaling components involved in eATP-promoted ion (Ca^2+^, K^+^) uptake, PG, and PTG were investigated. ATP stimulated while apyrase inhibited PG and PTG. Patch-clamping results showed that ATP promoted K^+^ and Ca^2+^ influx into pollen protoplasts. In loss-of-function mutants of P2K1 (*dorn1-1* and *dorn1-3*), heterotrimeric G protein α subunit (*gpa1-1*, *gpa1-2*), or cyclic nucleotide gated ion channel (*cngc2*, *cngc4*), eATP-stimulated PG, PTG, and ion influx were all impaired. Our results suggest that these signaling components may be involved in eATP-promoted PG and PTG by regulating Ca^2+^ or K^+^ influx in Arabidopsis pollen grains.

## 1. Introduction

The apoplast, including the cell wall and intercellular space, plays essential roles in modulating plant cell growth and development due to the existence of numerous signaling molecules within the apoplastic matrix. In recent years, growing evidence has shown that adenosine triphosphate (ATP) is present in the apoplast and participates in physiological processes such as vegetative growth, development, and stress response [1,2,3,4]. eATP is involved in maintaining cell viability [5] and the growth rate of cultured cells [6], regulating the rate or direction of the growth of roots [7], hypocotyls [8,9], and cotton fibers [10]. It is also involved in modulating stomatal movement [11,12,13]. Some stresses induce eATP release as a “danger signal”, which in turn induces resistance responses to disease, hypotonic conditions, high salinity, and cold stress [14,15,16,17,18].

The signal transduction of eATP has been intensively investigated over the past two decades. Two lectin receptor kinases, P2K1 and P2K2, were identified to be eATP receptors in *Arabidopsis thaliana* [19,20] involved in eATP-induced immune responses [13,14,21]. Signal transducers in the plasma membrane (PM), including heterotrimeric G proteins [12,22], NADPH oxidase [23,24], and Ca^2+^ channels [6,23,25,26], were reported to be involved in eATP signal transduction. Several secondary messengers, e.g., ROS, H_2_O_2_, NO, and Ca^2+^ may be responsible for the eATP-induced intracellular responses that eventually alter plant growth and development [6,17,26,27,28,29].

PG and PTG are fundamental processes in plant sexual reproduction. Ion intake is the main factor affecting the two processes [30,31,32]. Through the regulation of turgor pressure, K^+^ ions play essential roles in PG and PTG. Proper K^+^ channel expression is essential for Arabidopsis pollen hydration on stigma [33]. K^+^ influx causes pollen tube apex bursting and the release of sperm cells to complete double fertilization [34] and endows pollen with competitive ability [35]. Excessively low or high K^+^ or Ca^2+^ concentration inhibited PG and PTG in vitro in Arabidopsis [36,37]. Inward K^+^ channels that drive K^+^ absorption by pollen cells are involved in PG and PTG [33,38,39,40]. Ca^2+^ is involved in modulating the activity of some inward K^+^ channels in pollen protoplasts [39,41].

As a component of the cell wall and a cytoplasmic messenger, Ca^2+^ ions play essential roles in PG and PTG [31,32,42,43,44,45]. Sufficient Ca^2+^ is necessary for PG and PTG. Pollen tubes emerge from the aperture where high cytosolic Ca^2+^ concentration ([Ca^2+^]_cyt_) is localized [42,46,47]. [Ca^2+^]_cyt_ initiates pollen germination [46] and regulates the rate and direction of growth of pollen tubes [44,45,48,49,50,51]. Exocytosis occurs continuously during PG and PTG. Ca^2+^-regulated vesicle trafficking and fusion with the PM therefore play key roles in PG and PTG [51,52,53,54,55,56,57]. Ca^2+^ influx and signaling are responsible for stimuli-triggered or -regulated PG and PTG [58,59,60,61,62,63].

High concentrations of eATP show an inhibitory effect, while low concentrations of ATP show a promotive effect on PG and PTG [64,65,66]. We have reported that eATP stimulates the PG and PTG of *Nicotiana tabacum* by facilitating K^+^ or Ca^2+^ uptake [63]. Nevertheless, the universality of such an effect remains to be explored, and the signaling underlying eATP-stimulated PG and PTG remains unclear. Herein, we firstly investigated the response of pollen from 34 plant species to ATP addition and then elucidated the role of several signaling components in eATP-regulated PG and PTG of *Arabidopsis thaliana*.

## 2. Results

### 2.1. ATP Addition Impacts PG and PTG in Pollens from 34 Species

To verify the universality of eATP-regulated PG and PTG, we examined the effects of 0.1 mM ATP on PG and PTG of pollen grains from 34 species. Pollen germination rates increased significantly in 28 species (*p* < 0.05) and decreased significantly in 6 species (*p* < 0.05). Pollen tube length measurement results showed that tube growth was significantly promoted in 19 species (*p* < 0.05) and significantly inhibited in 5 species, while no effect was observed in 10 species (Figure 1).

### 2.2. eATP Regulates PG and PTG of Arabidopsis thaliana via K^+^ and Ca^2+^ Intake

To verify the role of ion uptake in eATP-promoted PG and PTG, *Arabidopsis thaliana* pollen was germinated in vitro, and the effects of apyrase or ATP addition on PG and PTG were investigated. In the basic medium, which contained 0.1 mM KCl, apyrase inhibited PG and PTG significantly. After 100 units/mL apyrase treatment, the pollen germination rate decreased from 39.2 ± 2.5% to 26.7 ± 3.4%, and the pollen tube length decreased from 16.9 ± 2.4 μm to 10.3 ± 1.8 μm (*p* < 0.05). Control treatment with heat-denatured apyrase or bovine serum albumin (BSA) did not affect PG or PTG (*p* > 0.05) (Figure 2A,D). At low K^+^ concentrations (0.02–0.1 mM), ATP supplementation (0.1 mM) promoted PG and PTG significantly, while at high K^+^ concentration (1.0 mM), ATP supplementation slightly inhibited PG but did not affect PTG (Figure 2B,E). To confirm that eATP supplementation promotes PG and PTG by increasing K^+^ uptake, 1 mM of the K^+^ channel blockers Ba^2+^, Cs^+^, or TEA (tetraethylammonium) was added into the basic medium containing 0.1 mM KCl. In these conditions, the effects of eATP on PG or PTG were significantly suppressed (*p* < 0.01) (Figure 2C,F).

To verify the role of Ca^2+^ in eATP-promoted PG and PTG, 0.1 mM ATP was added to medium containing EGTA, series of concentrations of Ca^2+^, or Ca^2+^ channel blockers. In 0.5 mM EGTA-containing medium, PG and PTG were significantly suppressed, and ATP supplementation did not have any effect. In medium containing 0.1 mM CaCl_2_, PG and PTG were both stimulated by 0.1 mM ATP. In medium containing 1 mM CaCl_2_, pollen tube growth was markedly stimulated, while pollen germination was suppressed by 0.1 mM ATP. In medium containing 0.1 mM Ca^2+^-permeable channel blockers (Gd^3+^ or La^3+^), resting PG and PTG were both suppressed, and added ATP did not have any effect on PG and PTG (Figure 3). These data indicate that eATP may promote PG and PTG by facilitating Ca^2+^ intake.

To verify the role of ATP hydrolytes in PG and PTG, 0.1 mM ADP, AMP, adenosine, adenine, or phosphate (Pi) was added to the medium containing 0.1 mM KCl or CaCl_2_. PG and PTG were not affected by these reagents in either KCl or CaCl_2_ medium (Appendix A). To verify the effect of various ATP salts on PG and PTG, 0.1 mM ATP sodium, Tris, or magnesium salt was added to the medium containing 0.1 mM KCl or CaCl_2_. PG and PTG were significantly promoted by the three ATP salts (*p* < 0.05) in both the KCl and CaCl_2_ media (Appendix A). To ensure that ATP acts as a signal rather than an energy carrier, 0.1 mM ATPγS, a weakly-hydrolyzable ATP analog, was added to the medium containing either 0.1 mM KCl or CaCl_2_. PG and PTG were significantly promoted by ATPγS (*p* < 0.05) in the KCl and CaCl_2_ media (Appendix A).

### 2.3. ATP Stimulates K^+^ and Ca^2+^ Influx in Arabidopsis thaliana Pollen Protoplast

To confirm that eATP stimulates K^+^ uptake in pollen grains, whole-cell patch clamping was performed to detect the effect of eATP on K^+^ conductance in the PM. In pollen protoplasts, a time-dependent inward-rectifying K^+^ current was recorded at a negative voltage of more than −140 mV. The addition of 1 mM CsCl significantly suppressed the current intensity, confirming that the inward currents may be carried by K^+^ influx (Figure 4A). The addition of 0.1 mM ATP stimulated the inward K^+^ current: the maximum current intensity increased from −129 ± 32 pA to −254 ± 34 pA at −200 mV (*n* = 7, *p* < 0.05), and the open potential shifted to more positive (Figure 4B). In CsCl-pretreated protoplasts, the effect of ATP on the inward K^+^ currents was markedly suppressed. The maximum current intensity decreased from −270 ± 35 pA to −83 ± 18 pA at −200 mV (*n* = 7, *p* < 0.05) (Figure 4C).

To further confirm the promotion of eATP on the Ca^2+^ uptake of pollen grains, the effect of 0.1 mM ATP on inward Ca^2+^ conductance was investigated. The addition of 0.1 mM ATP strongly promoted Ca^2+^ influx in pollen protoplasts: the maximum inward current intensity at −200 mV increased from −147.6 ± 21.9 pA to −243.1 ± 22.8 pA (*n* = 7, *p* < 0.05) (Figure 5A,B). GdCl_3_ (1 mM) significantly suppressed Ca^2+^ conductance (Figure 5A,C, *n* = 7, *p* < 0.05). In Gd^3+^-pretreated pollen protoplasts, ATP did not stimulate Ca^2+^ inward conductance (Figure 5A,C, *n* = 7, *p* > 0.05).

### 2.4. P2K1 Receptor, Heterotrimeric G Protein α Subunit and Two CNGCs Are Involved in eATP-Regulated PG and PTG by Modulating K^+^ and Ca^2+^ Influx

To verify the role of P2K1 receptor in eATP-regulated PG and PTG, pollen grains of two P2K1 receptor (DORN1) null mutants, *dorn1-1* and *dorn1-3*, were germinated in vitro. In 0.1 mM KCl-containing medium, 0.1 mM ATP addition did not affect the PG or PTG of *dorn1-1* nor *dorn1-3* mutants (Figure 6A,B). In 0.1 mM CaCl_2_-containing medium, ATP addition inhibited the PG of the two mutants but did not affect PTG in either (Figure 6E,F). In the pollen protoplasts of *dorn1-1* and *dorn1-3* mutants, K^+^ influx conductance was similar and exhibited weaker current intensity than wildtype pollen protoplasts. The addition of 0.1 mM ATP did not stimulate K^+^ conductance (Figure 6C,D). Ca^2+^ current intensity in pollen protoplasts of both mutants was weaker than that in wildtype, and the addition of 0.1 mM ATP did not stimulate Ca^2+^ conductance in either mutant (Figure 6G,H).

To verify the role of heterotrimeric G protein in eATP-regulated PG and PTG, pollen grains of two heterotrimeric Gα subunit null mutants, *gpa1-1* and *gpa1-2*, were germinated in vitro. In 0.1 mM KCl or CaCl_2_ containing medium, the addition of 0.1 mM ATP did not promote PG or PTG in *gpa1-1* and *gpa1-2* mutants (Figure 7A–F). In pollen protoplasts of *gpa1-1* and *gpa1-2* mutants, K^+^ influx conductance was similar to that of wildtype pollen protoplasts. The addition of 0.1 mM ATP did not stimulate K^+^ conductance. Ca^2+^ current intensity in the pollen protoplasts of both mutants was weaker than that in wildtype, and the addition of 0.1 mM ATP did not stimulate Ca^2+^ conductance in either mutant (Figure 7C–H).

To verify the role of CNGC in eATP-regulated PG and PTG, pollen grains of two CNGC null mutants, *cngc2* and *cngc4*, were germinated in vitro. In 0.1 mM KCl- or CaCl_2_-containing medium, the addition of 0.1 mM ATP did not stimulate PG or PTG in *cngc2* or *cngc4* mutants, while in 0.1 mM CaCl_2_-containing medium, PTG in *cngc2* mutants was suppressed by ATP (Figure 8A–F). In the pollen protoplasts of *cngc2* and *cngc4* mutants, K^+^ influx conductance was weaker than in wildtype pollen protoplasts, and the addition of 0.1 mM ATP did not stimulate K^+^ conductance. Ca^2+^ current intensity in pollen protoplasts of both mutants was weaker than in wildtype, and the addition of 0.1 mM ATP did not stimulate Ca^2+^ conductance in either mutant (Figure 8C–H).

## 3. Discussion

### 3.1. eATP Regulates PG and PTG in Dozens of Plant Species

eATP plays a role in vegetative and reproductive tissue cells in plants. The addition of mM levels of ATP inhibited PG and PTG [64,65], while low concentrations of eATP stimulated PG and PTG [63,66]. Here, the responses of 34 plant species in terms of PG and PTG pointed to a dual effect of eATP: eATP had a positive effect on the pollen of most plant species, while had an inhibitory effect on the pollen of a few other species. Such a dual regulatory effect of eATP had been noticed in various plant systems [2,15,67]. The dual effect might result from the sensitivity of pollen to eATP. In most species, PG and PTG were promoted or inhibited simultaneously. Nevertheless, the responses of PG and PTG to ATP were not consistent. PG in *Paeonia lactiflora*, *Magnolia denudata, Robinia pseudoacacia, Clivia miniata*, and *Pinus tabuliformis* was promoted by eATP, but PTG did not change markedly. PG in *Jasminum nudiflorum*, *Malus halliana*, and *Orychophragmus violaceus* was promoted by ATP, whereas PTG was inhibited. For *Kolkwitzia amabilis* and *Cotoneaster horizontalis*, PG was inhibited by ATP, but PTG was not affected. These results revealed the universality of eATP’s regulatory effect on PG and PTG. Further research into the mechanism of eATP-regulated PG or PTG will shed light on the mechanisms that regulate plant sexual reproduction.

### 3.2. ATP Regulates PG and PTG of Arabidopsis thaliana

It had been reported that eATP regulates PG and PTG in *Picea meyeri* [66] and *Nicotiana tabacum* [63]. However, null mutants of eATP signaling components are difficult to generate in these species. Here, Arabidopsis pollen grains were used to verify the role of eATP in PG and PTG, and several mutants of eATP signaling components were used to clarify the role of these molecules in PG, PTG, and ion influx.

The addition of apyrase inhibited PG and PTG (Figure 2), indicating that endogenous eATP may be involved in maintaining PG and PTG. Endogenous eATP may be secreted during pollen development. The addition of ATP stimulated PG and PTG, indicating that endogenous eATP might be not sufficient and that PG and PTG can be promoted by increasing eATP level. Our finding that the addition of high levels of ATP inhibited the PG and PTG of *Arabidopsis thaliana* is in agreement with previous reports [64,65]. These reports used a high concentration of ATP (>1 mM), compared to the lower concentration of 0.1 mM ATP added here, which stimulated PG and PTG. These contrasting results indicate that, in a certain concentration range, eATP acts as a positive regulator. ATP hydrolytes, including ADP, AMP, and adenosine, did not affect PG or PTG, indicating that ATP hydrolysis is unnecessary for this regulatory effect. The addition of ADP stimulated some reactions that are evoked by eATP [3,23,25,68,69]. In our previous work, ADP was found to promote the PG and PTG of tobacco [63]. The results presented here indicate that *Arabidopsis thaliana* pollen may respond to ATP differently than tobacco pollen. A weakly hydrolyzable ATP analog, ATPγS, promoted PG and PTG of *Arabidopsis thaliana* similarly to ATP, confirming that ATP may function in its intact form as a signaling molecule. Various ATP salts showed similar positive effects on PG and PTG (Appendix A), indicating that the cations in these salts did not affect PG or PTG.

### 3.3. K^+^/Ca^2+^ Influx Mediates ATP Regulation of PG and PTG

In our previous work, we demonstrated the existence of eATP-regulated K^+^ and Ca^2+^ uptake in tobacco pollen [63]. Here, we found that K^+^ and Ca^2+^ uptake is necessary for resting or eATP-stimulated PG and PTG in *Arabidopsis thaliana*. Similar to the results in tobacco pollen, we found that ATP promoted PG and PTG in medium containing low concentrations of K^+^ or Ca^2+^ and inhibited PG and PTG in medium containing high concentrations of K^+^ or Ca^2+^. In medium containing 1 mM CaCl_2_, the pollen tube growth of tobacco was inhibited, while the pollen tube growth of *Arabidopsis* was strongly promoted (Figure 3B). In tobacco pollen, 0.1 mM Ca^2+^ channel blockers weakly suppressed eATP-promoted ion influx and PG and PTG, while in *Arabidopsis thaliana*, 0.1 mM Ca^2+^ channel blockers almost entirely blocked ATP-induced Ca^2+^ influx and PG and PTG. These results indicate that K^+^ and Ca^2+^ influx through ion channels may be a central event in PG and PTG, that endogenous eATP is necessary for maintaining resting ion uptake, that apyrase decreases eATP levels, and that decreased ion influx might explain apyrase-inhibited PG and PTG. The positive effects of ATP-supplementation on PG and PTG may partially result from increased ion absorption. The proper influx and accumulation of ions is essential for PG and PTG, while the overload of ions suppresses PG and PTG [32,33,37,40,45]. The data presented here suggest that, when eATP or ion concentrations are above normal levels, the overload of K^+^ or/and Ca^2+^ influx may inhibit PG and PTG.

### 3.4. Signaling Underlying ATP-Regulated PG and PTG of Arabidopsis thaliana

P2K1 receptor DORN1 was the first reported eATP receptor and is involved in plant immune and stress responses [13,14,21]. However, the role of the P2K1 receptor in sexual reproduction has not been investigated, although DORN1 is expressed in flowers (Appendix A). Results here showed that DORN1 is involved in PG and PTG. In 0.1 mM KCl-containing medium, the resting pollen germination rate and pollen tube length were similar between wildtype and both *DORN1*-null mutants, while in 0.1 mM CaCl_2_-containing medium, the pollen germination rate of both mutants was significantly lower than in wildtype, indicating that DORN1 is necessary for maintaining Ca^2+^ uptake and resting pollen germination, possibly through interactions with endogenous eATP. Patch clamping showed that resting K^+^ and Ca^2+^ current intensities were weaker than those in wildtype, demonstrating a role of DORN1 in maintaining the resting ion influx. Added ATP did not promote PG, PTG, or ion influx in *dorn1-1* and *dorn1-3*, indicating that DORN1 plays a key role in ATP-regulated PG and PTG through regulating K^+^/Ca^2+^ influx in pollen cells. DORN1 is involved in eATP-regulated intracellular Ca^2+^ concentration elevation in Arabidopsis root cells [19,26,70,71]. The results here provided new evidence for a role for DORN1 in eATP-evoked Ca^2+^ signaling in reproductive cells as well. Heterotrimer G protein is involved in eATP-regulated root growth and stomatal movement and acts by regulating Ca^2+^ influx [3,12,22,72,73]. Heterotrimeric G protein-regulated PG, PTG, and Ca^2+^ influx in *Arabidopsis thaliana* was reported in our previous work [61]. Nevertheless, the upstream regulator that governs the heterotrimeric G protein is still unknown. The data presented here indicates that eATP may be an upstream regulator of heterotrimeric G protein in pollen cells, suggesting that eATP-stimulated G protein activation may play similar roles in vegetable and reproductive growth. The resting pollen germination rate and pollen tube length of *gpa1-1* and *gpa1-2* mutants were lower than those in wildtype, and the addition of ATP did not promote PG or PTG in either mutant, indicating that Gα may play a key role in eATP-regulated PG and PTG. In the protoplasts of both null mutants, resting K^+^ or Ca^2+^ current intensities were slightly weaker than those in wildtype, and the addition of ATP did not stimulate K^+^ or Ca^2+^ conductance, indicating that Gα may be involved in eATP-regulated ion influx. These data provide new evidence that heterotrimeric G protein is involved in eATP-regulated PG and PTG through regulating K^+^ or Ca^2+^ influx.

As channels permeable to divalent cations and K^+^, CNGCs had been reported to participate in key processes involved in plant growth, development, and stress responses, including apical growth, the vegetative development of plants, sexual reproduction, immunity, response to drought, and temperature stress [26,74,75,76,77]. CNGC2 and CNGC4 form ion channels by heterodimerization in *Arabidopsis thaliana*, after which they mediate ion influx and function in plant growth and pathogen defense [78,79,80]. In our preliminary experiments, the responses of pollen from several *CNGC* null mutants to ATP supplementation were investigated. Most mutants responded to eATP similarly to wildtype (data not shown). Only mutants for CNGC2 and CNGC4 responded to eATP different from Col-0 pollen. The resting pollen germination rates of *cngc2* or *cngc4* mutants were lower than that of wildtype, indicating that ion channels containing CNGC2 and CNGC4 are essential for maintaining Ca^2+^ and K^+^ uptake. In pollen grains from both mutants, the addition of ATP did not promote PG or PTG, indicating that CNGC2 and CNGC4 may play key roles in eATP-stimulated PG and PTG. Patch clamping data showed that resting K^+^ and Ca^2+^ conductance in mutants was weaker than in wildtype, confirming that resting ion influx may be mediated by ion channels containing CNGC2 and CNGC4. The addition of ATP did not stimulate K^+^ or Ca^2+^ conductance in the pollen protoplasts of *cngc2* or *cngc4* mutants, indicating that eATP-stimulated ion influx may be mediated by CNGC2/CNGC4-containing ion channels. Single null mutants of CNGC2 or CNGC4 showed impaired resting and eATP-stimulated ion influx, indicating that the heterodimerization of these two proteins is necessary for the construction of functional ion channels. It was reported that CNGC18 and CNGC16 are involved in the pollen development, germination and tube growth in *Arabidopsis thaliana* [59,81,82], whether these CNGCs are involved in eATP signaling need to be further investigated.

Our results confirmed which components are involved in eATP-stimulated PG, PTG, and ion flux and provided new clues to help elucidate the mechanisms of eATP-regulated reproduction. Based on these results, it is reasonable to speculate that eATP first binds the P2K1 receptor, then stimulates heterotrimer G protein, then activated Gα stimulates CNGC2/CNGC4, finally leading to increased Ca^2+^/K^+^ influx and the promotion of PG and PTG. How these components are connected remains unclear. Some receptor-like kinases directly bind and phosphorylate heterotrimeric G protein or activate heterotrimeric G protein through RGS1 (regulator of G protein signaling 1) [83,84,85]. Hence, the interaction of DORN1 and RGS1 (or heterotrimeric Gα) need to be carefully investigated before we can draw a conclusion. Heterotrimeric G protein is involved in the regulation of ion flux in plant cells [86]; however, whether heterotrimeric G protein directly activates ion channels or does so indirectly through altering membrane potential or generating cytoplasmic messengers (e.g., ROS or NO) needs to be clarified in the future.

## 4. Materials and Methods

### 4.1. Plant Materials

Pollen grains of 34 plant species (Table 1) were collected from newly opened flowers and then stored (adding silica gel as a dehydrator) at −20 °C.

*Arabidopsis thaliana* plants were grown in a greenhouse running at 22 °C with illumination intensity of 120–130 μmol·m^−2^·s^−1^, relative humidity of 70%, and a 16/8 h light/dark cycle.

### 4.2. In Vitro Pollen Germination

The basic medium contained 18% sucrose and had a pH of 7.0. Nutrient salts were added according to individual experimental aims. All reagents were dissolved in de-ionized water, and 0.5% agarose was added to solidify the medium for pollen germination.

Pollen grains were suspended in a basic medium and spread onto the surface of a solid medium. Then, pollen grains were incubated in the dark at 28 °C with a relative humidity of 100%. The incubation duration varied among different species since the time needed to reach maximum germination rate for each species is different (see detail in Table 1). After incubation, pollen grains were photographed with a DS-Fi1c imaging system equipped to a microscope (Nikon Eclipse 80i, Japan). Images were analyzed with NIS-Element BR 4.0 software to count germinated pollen grains and measure pollen tube length. In each experiment, no less than 300 pollen grains were counted to obtain germination rate (pollen grains with emerging tubes longer than 5 μm were considered as germinated pollen), and the lengths of no less than 150 pollen tubes were measured. Data from 3 replicates were calculated to obtain mean values. Data was analyzed with Statistica 8.0 software.

### 4.3. Protoplast Isolation

Pollen grains of *Arabidopsis thaliana* were incubated at 28 °C in a bath solution (20 mM CaCl_2_, 5 mM MES, 1.5 M sorbitol, pH 5.8), which contained 1.5% cellulase (Onozuka RS, Yakult, Japan) and 1% macerozyme (R-10, Yakult, Japan). After 1.5 h, the isolated pollen protoplasts were centrifuged and washed with the bath solution twice.

### 4.4. Patch-Clamp Recording

All patch-clamp solutions were adjusted to 1.5 Osm/kg with sorbitol. For K^+^ current detection, the bath solution specifications were 10 mM KC_6_H_11_O_7_, 5 mM Mes, and pH 5.8., and the pipette solution specifications were 100 mM KC_6_H_11_O_7_, 0.1 mM CaCl_2_, 2 mM MgCl_2_, 10 mM Hepes, 2 mM Mg·ATP, 1.1 mM EGTA, 4 mM KOH, and pH 7.2. For Ca^2+^ current detection, the bath solution specifications were 20 mM CaCl_2_, 5 mM Mes, and pH 5.8, and the pipette solution specifications were 0.5 mM CaCl_2_, 8.5 mM Ca(OH)_2_, 2 mM Mg∙ATP, 0.5 mM ATP∙Tris, 10 mM BAPTA, 15 mM Hepes, and pH 7.2.

Patch-clamp recording methods were manipulated according to Véry & Davies [87]. Glass pipettes (World Precision Instruments, FL, USA) were pulled with a vertical puller (PC-10, Narishige, Japan). A whole-cell voltage clamping configuration was used. Current signal was recorded with an AXON 200B amplifier (AXON Instruments, IL, USA) equipped with Digidata 1440A data acquisition system (AXON Instruments, IL, USA), which was controlled by Clampex 10.3 software (AXON Instruments, IL, USA). Data was sampled at 1 kHz and filtered at 200 Hz. Voltage-clamp protocols included a series of depolarizing and/or hyperpolarizing steps of 1.5 s from a holding potential in the range of 0 to −200 mV. Current–voltage relationships (I-V curves) were then constructed with total whole-cell currents measured after 1.5 s step-voltage clamping. Data were analyzed with Microcal Origin 6.0.

## Figures and Tables

**Figure 1 plants-10-01743-f001:**
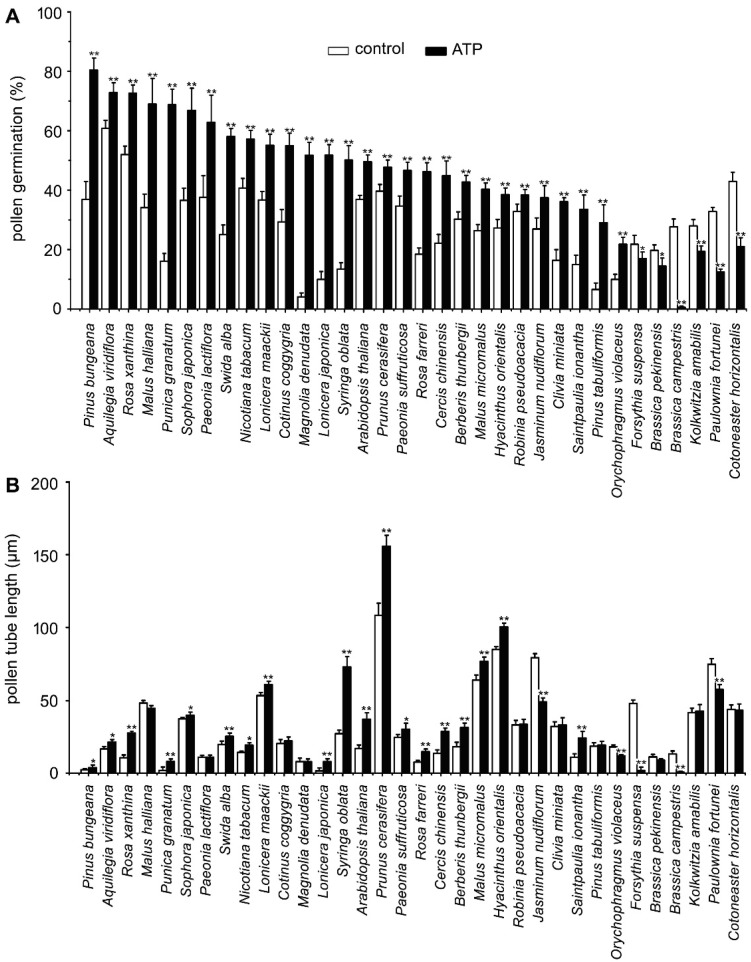
ATP-regulated PG and PTG in various plant species. Pollen grains of 34 plant species were germinated in vitro in the medium containing 18% sucrose, 0.1 mM KCl, and 0.5% agarose. Germination duration of each species is listed in Table 1. PG (**A**) and PTG (**B**) were noted. In each experiment, at least 300 pollen grains or 150 pollen tubes were counted or measured. Data from 3 replicates were calculated to obtain mean ± SD. Student’s *t*-test *p* values: * *p* < 0.05, ** *p* < 0.01.

**Figure 2 plants-10-01743-f002:**
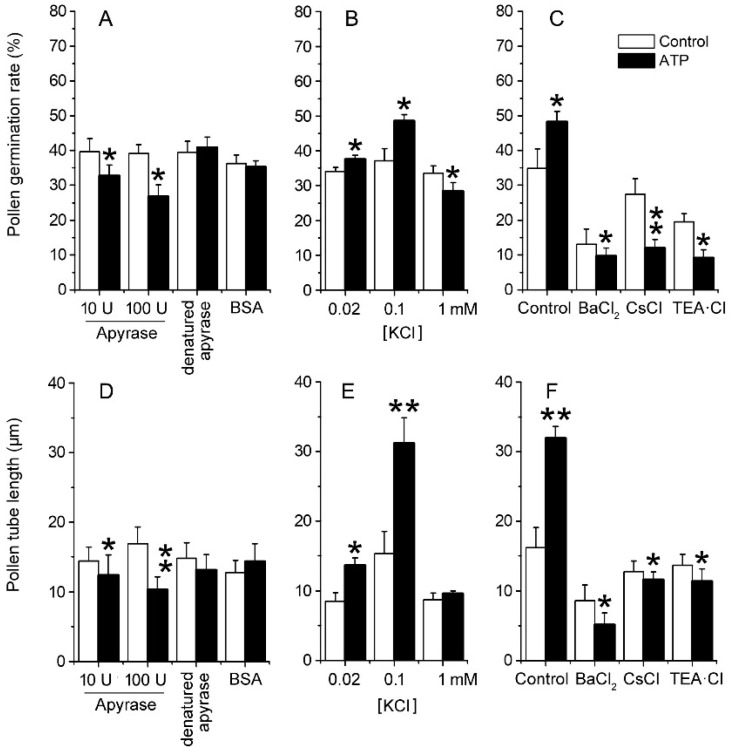
eATP regulates the PG and PTG of *Arabidopsis thaliana* by facilitating K^+^ uptake. Apyrase-inhibited PG (**A**) and PTG (**D**), ATP-regulated PG (**B**), and PTG (**E**) in medium containing serial concentrations of KCl. K^+^ channel blockers (1 mM) inhibited ATP-promoted PG (**C**) and PTG (**F**). In (**A**–**F**), the K^+^ concentration in the medium is 0.1 mM. In each experiment, at least 300 pollen grains were counted to obtain PG, and at least 150 pollen tubes were measured to obtain the pollen tube length. Data from 3 replicates were combined to obtain the mean ± SD. Student’s *t* test *p* values: * *p* < 0.05, ** *p* < 0.01.

**Figure 3 plants-10-01743-f003:**
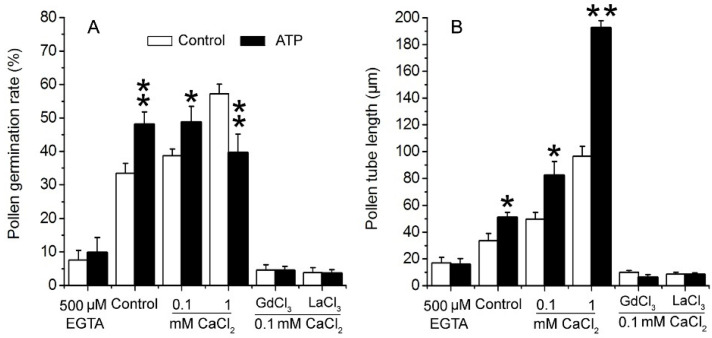
eATP regulates PG and PTG of *Arabidopsis thaliana* by facilitating Ca^2+^ uptake. ATP (0.1 mM) regulates PG (**A**) and PTG (**B**) in medium containing a Ca^2+^ chelator, serial concentrations of Ca^2+^, and channel blockers (0.1 mM). In each experiment, at least 300 pollen grains were counted to obtain PG, and at least 150 pollen tubes were measured to obtain the pollen tube length. Data from 3 replicates were combined to obtain the mean ± SD. Student’s *t* test *p* values: * *p* < 0.05, ** *p* < 0.01.

**Figure 4 plants-10-01743-f004:**
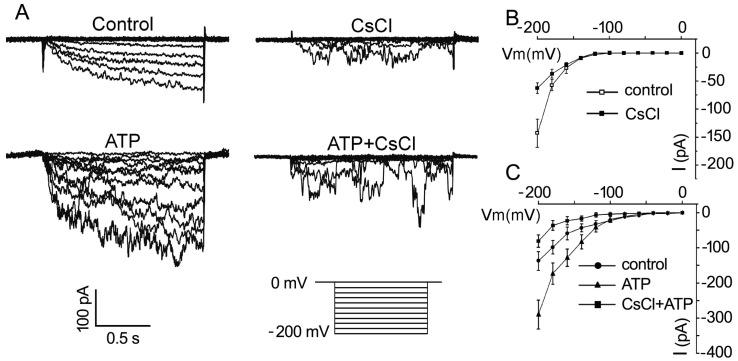
CsCl inhibits eATP-stimulated K^+^ influx in the protoplasts of *Arabidopsis* pollen grains. (**A**) CsCl (1 mM) inhibited resting and 0.1 mM ATP-stimulated inward K^+^ currents. Representative current traces from step-voltage clamping are shown. (**B**,**C**) I-V relationship curves of inward K^+^ currents (*n* = 7). In each experiment, data from 7 protoplasts were recorded and combined to obtain the mean ± SD.

**Figure 5 plants-10-01743-f005:**
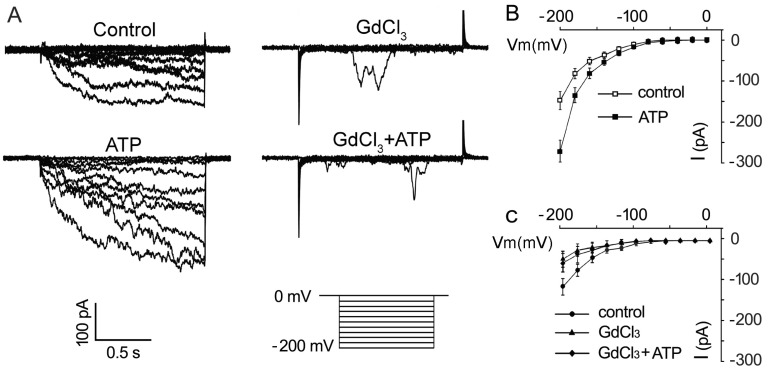
GdCl_3_ inhibits eATP-stimulated Ca^2+^ influx in the protoplasts of *Arabidopsis* pollen grain. (**A**) GdCl_3_ (0.1 mM) inhibited resting and 0.1 mM ATP-stimulated inward Ca^2+^ currents. Representative current traces from step-voltage clamping are shown. (**B**,**C**) I-V relationship curves of the inward Ca^2+^ currents (*n* = 7). In each experiment, data from 7 protoplasts were recorded and combined to obtain the mean ± SD.

**Figure 6 plants-10-01743-f006:**
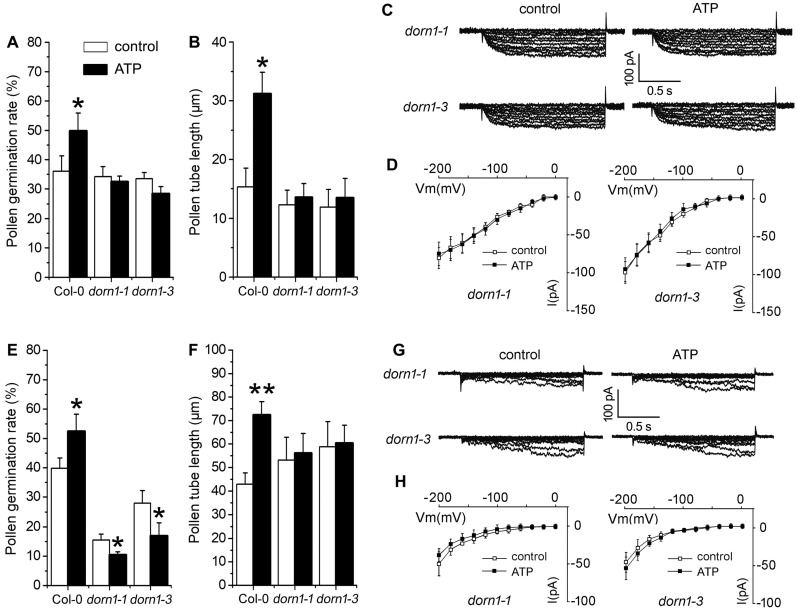
P2K1 receptor (DORN1) is involved in the eATP-regulated PG, PTG, and ion influx. Two *DORN1* null mutants (*dorn1-1*, *dorn1-3*) were used. (**A**–**D**) show the effects of ATP on PG (**A**) and PTG (**B**) in medium containing 0.1 mM KCl and the corresponding K^+^ influx current traces (**C**) and I-V relationship curves ((**D**), *n* = 7). (**E**–**H**) show the effect of ATP on PG (**E**) and PTG (**F**) in medium containing 0.1 mM CaCl_2_ and the corresponding Ca^2+^ influx current traces (**G**) and I-V relationship curves ((**H**), *n* = 7). In the pollen germination experiment, at least 300 pollen grains or 150 pollen were counted or measured; data from 3 replicates were combined to obtain the mean ± SD. Student’s *t* test *p* values: * *p* < 0.05, ** *p* < 0.01. In the patch clamp experiment, data from 7 pollen protoplasts were recorded and combined to obtain the mean ± SD.

**Figure 7 plants-10-01743-f007:**
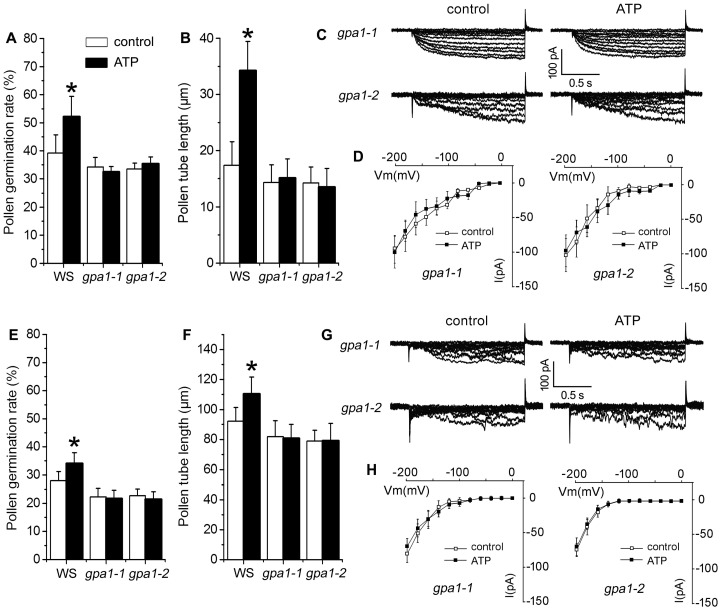
Heterotrimeric G protein α subunit (GPA1) is involved in eATP-regulated PG, PTG, and ion influx. Two *GPA1* null mutants (*gpa1-1*, *gpa1-2*) were used. (**A**–**D**) show the effects of ATP on PG (**A**) and PTG (**B**) in medium containing 0.1 mM KCl with corresponding K^+^ influx current traces (**C**) and I-V relationship curves ((**D**), *n* = 7). E–H show the effects of ATP on PG (**E**) and PTG (**F**) in medium containing 0.1 mM CaCl_2_ with corresponding Ca^2+^ influx current traces (**G**) and I-V relationship curves ((**H**), *n* = 7). In the pollen germination experiment, at least 300 pollen grains or 150 pollen were counted or measured; data from 3 replicates were combined to obtain the mean ± SD. Student’s *t* test *p* values: * *p* < 0.05. In the patch clamp experiment, data from at least 7 pollen protoplasts were recorded and combined to obtain the mean ± SD.

**Figure 8 plants-10-01743-f008:**
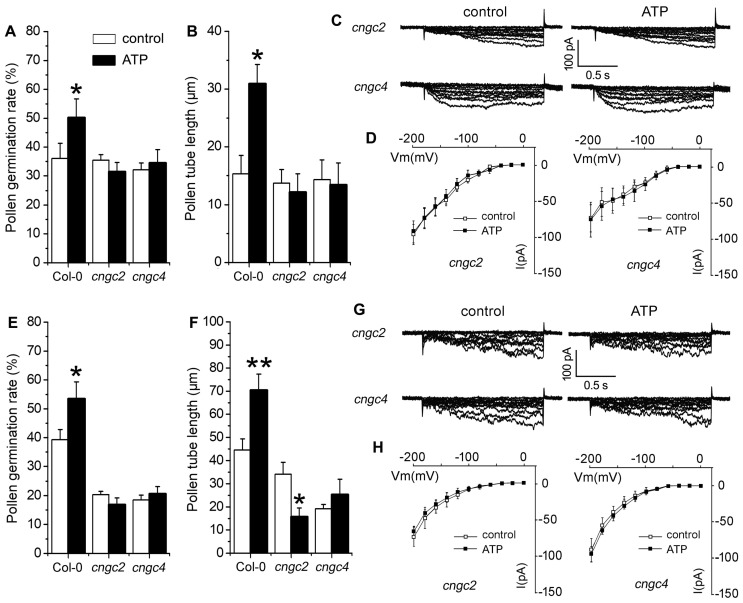
CNGC2 and CNGC4 are involved in eATP-regulated PG, PTG, and ion influx. Two *CNGCs* null mutants (*cngc2*, *cngc4*) were used. (**A**–**D**) show the effects of ATP on PG (**A**) and PTG (**B**) in medium containing 0.1 mM KCl with corresponding K^+^ influx current traces (**C**) and I-V relationship curves ((**D**), *n* = 7). (**E**–**H**) show the effects of ATP on PG (**E**) and PTG (**F**) in medium containing 0.1 mM CaCl_2_ with corresponding Ca^2+^ influx current traces (**G**) and I-V relationship curves ((**H**), *n* = 7). In the pollen germination experiment, at least 300 pollen grains or 150 pollen were counted or measured; data from 3 replicates were combined to obtain the mean ± SD. Student’s *t* test *p* values: * *p* < 0.05, ** *p* < 0.01. In the patch clamp experiment, data from at least 7 pollen protoplasts were recorded and combined to obtain the mean ± SD.

**Table 1 plants-10-01743-t001:** Plant materials for in vitro pollen germination (sorted by the initials of the Latin names of plant species).

	Latin Name	Family	Class	Phylum	Germination Duration (min)
1	*Aquilegia viridiflora*	Ranunculaceae	Dicotyledoneae	Angiospermae	30
2	*Arabidopsis thaliana*	Brassicaceae	Dicotyledoneae	Angiospermae	300
3	*Berberis thunbergii*	Berberidaceae	Dicotyledoneae	Angiospermae	40
4	*Brassica campestris*	Brassicaceae	Dicotyledoneae	Angiospermae	240
5	*Brassica pekinensis*	Brassicaceae	Dicotyledoneae	Angiospermae	150
6	*Cercis chinensis*	Leguminosae	Dicotyledoneae	Angiospermae	90
7	*Clivia miniata*	Amaryllidaceae	Monocotyledoneae	Angiospermae	420
8	*Cotinus coggygria*	Anacardiaceae	Dicotyledoneae	Angiospermae	40
9	*Cotoneaster horizontalis*	Rosaceae	Dicotyledoneae	Angiospermae	60
10	*Forsythia suspensa*	Oleaceae	Dicotyledoneae	Angiospermae	120
11	*Hyacinthus orientalis*	Hyacinthaceae	Monocotyledoneae	Angiospermae	80
12	*Jasminum nudiflorum*	Oleaceae	Dicotyledoneae	Angiospermae	90
13	*Kolkwitzia amabilis*	Caprifoliaceae	Dicotyledoneae	Angiospermae	60
14	*Lonicera maackii*	Caprifoliaceae	Dicotyledoneae	Angiospermae	60
15	*Lonicera japonica*	Caprifoliaceae	Dicotyledoneae	Angiospermae	120
16	*Magnolia denudata*	Magnoliaceae	Dicotyledoneae	Angiospermae	2880
17	*Malus halliana*	Rosaceae	Dicotyledoneae	Angiospermae	30
18	*Malus micromalus*	Rosaceae	Dicotyledoneae	Angiospermae	60
19	*Nicotiana tabacum*	Solanaceae	Dicotyledoneae	Angiospermae	30
20	*Orychophragmus violaceus*	Brassicaceae	Dicotyledoneae	Angiospermae	180
21	*Paeonia suffruticosa*	Paeoniaceae	Dicotyledoneae	Angiospermae	60
22	*Paeonia lactiflora*	Paeoniaceae	Dicotyledoneae	Angiospermae	60
23	*Paulownia fortunei*	Scrophulariaceae	Dicotyledoneae	Angiospermae	60
24	*Pinus bungeana*	Pinaceae	Coniferopsida	Gymnosperm	4320
25	*Pinus tabulaeformis*	Pinaceae	Coniferopsida	Gymnosperm	4320
26	*Punica granatum*	Punicaceae	Dicotyledoneae	Angiospermae	120
27	*Prunus cerasifera*	Rosaceae	Dicotyledoneae	Angiospermae	60
28	*Rosa xanthina*	Rosaceae	Dicotyledoneae	Angiospermae	20
29	*Robinia pseudoacacia*	Leguminosae	Dicotyledoneae	Angiospermae	30
30	*Rosa farreri*	Rosaceae	Dicotyledoneae	Angiospermae	30
31	*Saintpaulia ionantha*	Gesneriaceae	Dicotyledoneae	Angiospermae	180
32	*Sophora japonica*	Leguminosae	Dicotyledoneae	Angiospermae	30
33	*Swida alba*	Corneceae	Dicotyledoneae	Angiospermae	60
34	*Syringa oblata*	Oleaceae	Dicotyledoneae	Angiospermae	90

## Data Availability

The study did not report any data.

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
