# Peer review of "P2K1 Receptor, Heterotrimeric Gα Protein and CNGC2/4 Are Involved in Extracellular ATP-Promoted Ion Influx in the Pollen of Arabidopsis thaliana"

_plants, 2021, doi:10.3390/plants10081743_

Round 1

Reviewer 1 Report

It is interesting to see ATP involved in signalling rather than being an energy provider.

The experimental methods are well designed with good blend of molecular biology and electrophysiology.

The large variation in response (positive, negative or none) of different plant species to eATP is interesting. However, the heading “3.1. eATP Universally Regulates PG and PTG in Dozens of Plant Species.” is not quite justified: the effect of eATP is not universal.

There are some small English expression problems:

Line 49: “By effectively regulate turgor pressure, K+ channels…” to “By effective regulation of turgor pressure”

Line 110: “serial concentrations” to “series of concentrations”

Having the sentence “this is a figure” in front of each figure caption is unnecessary. Similarly with “this is a table” for tables.

Reviewer 2 Report

The regulation of pollen tube growth by ATP is very interesting. In this study, Wu et al., report that P2K1 receptor, heterotrimeric G protein and CNGC2/4 are involved in extracellular ATP-promoted ion influx in pollen. This deepens our understanding of the molecular mechanisms by which eATP regulates pollen tube growth. On the whole, this study is good, I have some suggestions for improving the quality of the article.

  1. Change the phrase "pollen cell" to" pollen "in the title.
  2. There is a lack of scale in Fig6C and G, Fig 7C and G, Fig 8C and G. It is recommended that the scale of the figure should be made according to Figure 4A.
  3. CNGC18 has the highest expression level in Arabidopsis pollen and plays a critical role for pollen tube growth.Whether it is involved in the regulation of calcium and potassium currents in pollen plasma membrane by eATP needs to be analyzed or discussed.
  4. Does DORN1indeed expressed in pollen should be confirmed or cite established literature.
  5. The discussion part of the manuscriptneeds strengthen, such as the relationship between P2K1 receptor, heterotrimeric Gα protein and CNGC2/4.

Author Response

This manuscript is a resubmission of an earlier submission. The following is a list of the peer review reports and author responses from that submission.